# Molecular Interactions between Neuroglobin and Cytochrome *c:* Possible Mechanisms of Antiapoptotic Defense in Neuronal Cells

**DOI:** 10.3390/biom13081233

**Published:** 2023-08-10

**Authors:** Marina A. Semenova, Rita V. Chertkova, Mikhail P. Kirpichnikov, Dmitry A. Dolgikh

**Affiliations:** 1Shemyakin-Ovchinnikov Institute of Bioorganic Chemistry, Russian Academy of Sciences, Miklukho–Maklaya St. 16/10, 117997 Moscow, Russia; 2Biology Department, Lomonosov Moscow State University, Leninskie Gory, 119899 Moscow, Russia

**Keywords:** neuroglobin, neuroprotection, cytochrome *c*, heme proteins, protein–protein interaction, apoptosis, globins

## Abstract

Neuroglobin, which is a heme protein from the globin family that is predominantly expressed in nervous tissue, can promote a neuronal survivor. However, the molecular mechanisms underlying the neuroprotective function of Ngb remain poorly understood to this day. The interactions between neuroglobin and mitochondrial cytochrome *c* may serve as at least one of the mechanisms of neuroglobin-mediated neuroprotection. Interestingly, neuroglobin and cytochrome *c* possibly can interact with or without electron transfer both in the cytoplasm and within the mitochondria. This review provides a general picture of molecular interactions between neuroglobin and cytochrome *c* based on the recent experimental and computational work on neuroglobin and cytochrome *c* interactions.

## 1. Introduction

Neuroglobin (Ngb) is a six-coordinate heme protein from the globin family that is predominantly expressed in nervous tissue [1]. Intracellular Ngb concentrations range from 100 μM in the neurons of the hypothalamus and the retinal cells to 1 μM in the other neurons [1,2,3]. Furthermore, there is an average 2-5-fold increase in *NGB* gene expression in response to various stress signals (e.g., hypoxia or oxidative stress) [4,5]. Ngb is a cytoplasmic protein, although there is evidence for Ngb localization not only in close proximity to mitochondria, but also within mitochondria [6,7,8]. However, Ngb lacks a mitochondrial target sequence, and the exact mechanism regulating Ngb mitochondrial localization is still unclear [9].

Ngb is a monomer with a molecular weight of about 17 kDa, consists of eight α-helices (A-H), and has a typical three-over-three globin fold [1,10,11]. Ngb contains one heme group, and, in the absence of exogenous ligands, the axial coordination positions of the heme are occupied by two histidine residues (His64-Fe-His96) [11,12]. Human Ngb features three cysteine residues (Cys46, Cys55, and Cys120), two of which form an intramolecular disulfide bridge (Cys46–Cys55) [1,10,11]. The formation of other intramolecular disulfide bridges in Ngb involving Cys120 and Cys46 or Cys55 is considered to be sterically impossible [11,13]. The significance of the Cys46–Cys55 disulfide bridge is determined by its effect on the conformation of the CD-loop (the region between α-helices C and D) and modulation of Ngb functionality. Upon cleavage of this disulfide bridge, the bond between the heme iron and His64 is strengthened, thus resulting in a ten-fold decrease of the His64 dissociation rate. As a consequence, the affinity of Ngb for exogenous ligands is reduced because the dissociation of His64 from the heme is a necessary and rate-determining step for the binding of exogenous ligands to Ngb [11,14]. However, it is suggested that, in vivo, more than 88% of Ngb molecules are in the ferrous deoxy form without a disulfide bridge [15,16,17].

Similar to hemoglobin and myoglobin, Ngb reversibly binds O_2_ and other small gaseous ligands [12,18]. Ngb catalyzes the NO deoxygenation reaction, thus yielding NO_3_^−^ [19,20], as well as NO generation from NO_2_^−^ (nitrite reductase activity) [21,22,23].

Ngb has been shown to promote neuronal survival in conditions such as ischemia, hypoxia, Alzheimer’s and Huntington’s diseases, oxidative stress, stroke, spinal cord injury, retinal degeneration, arsenic poisoning, etc., in numerous in vitro and in vivo studies [4,5,9,24,25,26,27,28,29]. However, the molecular mechanisms underlying the neuroprotective function of Ngb remain poorly understood to this day. Because Ngb can bind O_2_, it was initially assumed that Ngb, like myoglobin, acts as an O_2_ depot and transporter under hypoxia [1,2,3]. However, relatively low cellular concentrations of Ngb and a very rapid autoxidation contradict this assumption [30].

Recent hypotheses suggest that Ngb-mediated neuroprotection is based on its involvement in various biochemical cascades of the cell [4,9,24,28,29]. Ngb can detoxify reactive oxygen and nitrogen species [31,32,33,34], and there is evidence of Ngb involvement in the Wnt/β-catenin pathway [35,36], as well as in the PI3K/Akt/MAPK signaling pathway [37,38]. Additionally, the protein–protein interactions between Ngb and voltage-dependent anion channels (VDACs) [6,7,39], α-subunits of heterotrimeric G-protein [40,41,42], and mitochondrial cytochrome *c* (Cyt *c*) [30,43] may also contribute to Ngb-mediated neuroprotection. This review is focused on the molecular interactions between Ngb and Cyt *c*.

Mitochondrial Cyt *c*, a multifunctional heme protein, plays a crucial role as an electron transporter in the respiratory chain. During the activation of apoptosis, Cyt *c* translocates from the mitochondria into the cytoplasm, where it amplifies the external apoptotic signal or initiates the caspase cascade through the intrinsic pathway (making it Cyt *c*-dependent) [44,45,46,47]. In addition to physiological role of apoptosis in nervous tissue, it is also associated with pathological conditions, such as neurodegenerative diseases, strokes, ischemia, hypoxia, etc. Furthermore, the initiation of apoptosis occurs under the influence of various stress stimuli, including pathogens, radiation, chemotherapeutic drugs, oxidative stress, etc., which cause mitochondrial dysfunction [48,49,50]. 

The key event of the intrinsic apoptotic pathway is the outer mitochondrial membrane permeabilization, which leads to the release of various proapoptotic factors, including Cyt *c*, into the cytoplasm. Furthermore, the activation of the caspase cascade depends on the efficiency of Cyt *c* binding to apoptotic protease activating factor-1 (Apaf-1), thereby leading to apoptosome assembly. It should be noted that only ferric Cyt *c* can bind to Apaf-1, thereby activating apoptosome assembly and triggering the caspase cascade [51].

The hypothesis about the interaction between Ngb and Cyt *c* was proposed based on the kinetic studies of the redox reaction between Ngb(Fe^2+^) and Cyt *c*(Fe^3+^):Ngb(Fe^2+^) + Cyt *c*(Fe^3+^) → Ngb(Fe^3+^) + Cyt *c*(Fe^2+^).(1)
This reaction was initially observed by absorbance spectroscopy under anaerobic conditions [43]. According to this hypothesis, Ngb(Fe^2+^) reacts with Cyt *c*(Fe^3+^) when the latter translocates into the cytoplasm during the activation of the intrinsic apoptotic pathway. This reaction prevents Cyt *c*(Fe^3+^) from interacting with Apaf-1 and, thus, prevents apoptosome assembly and ultimately blocks the initiation of apoptosis [30,43,52]. Later, it was hypothesized that Ngb can potentially interact with Cyt *c*(Fe^3+^), regardless of its redox state [53,54].

Therefore, Ngb can protect neurons from apoptosis by reducing the concentration of cytosolic Cyt *c*(Fe^3+^), which can increase under conditions that cause mitochondrial dysfunction and during normal mitochondrial function (reducing “accidental” Cyt *c* release) [30].

The aim of this review is to summarize the existing data and provide a general overview of the molecular interactions between Ngb and Cyt *c* that underlie at least one of the mechanisms of Ngb-mediated neuroprotection. 

## 2. Electron Transfer between Ferrous Neuroglobin and Ferric Cytochrome *c*

The investigation of interactions between Ngb and Cyt *c* began with the observation of the very rapid electron transfer from recombinant murine Ngb(Fe^2+^) to Cyt *c*(Fe^3+^) [43]. The second-order rate constant for the reaction (k = 2 × 10^7^ M^−1^ s^−1^) is close to the known physiologically significant Cyt *c* redox reactions rate constants, e.g., the reduction of cytochrome *c* oxidase or oxidation of cytochrome *b5* [43,46,55]. It should be noted that, unlike human Ngb, murine Ngb lacks the disulfide bridge due to the replacement of Cys46 by Gly [1]. Hence, it can be concluded that the presence of the Ngb disulfide bridge does not affect electron transfer from Ngb(Fe^2+^) to Cyt *c*(Fe^3+^) [43]. 

However, a recent study [56] has come to somewhat controversial conclusions. The electron transfer between recombinant human Ngb(Fe^2+^) and Cyt *c*(Fe^3+^) was studied using a nanoporous gold electrode. The immobilization of Ngb onto the electrode surface enabled the rapid reduction of the Ngb heme, after which Cyt *c*(Fe^3+^) was added to the solution, and its reduction to Cyt *c*(Fe^2+^) was observed. In the case of the Ngb C55S mutant without the disulfide bridge (Cys46–Cys55), there were almost no voltammogram changes upon Cyt *c*(Fe^3+^) addition. Thus, it was concluded that the disulfide bridge has significance for electron transfer between Ngb(Fe^2+^) and Cyt *c*(Fe^3+^). These results are inconsistent with the previously obtained data on electron transfer between murine Ngb(Fe^2+^) and Cyt *c*(Fe^3+^) [43]. It is assumed that the Ngb disulfide bridge formation under oxidative stress conditions is potentially capable of modulating the functionality of Ngb. For instance, we can consider the electron transfer to Cyt *c*(Fe^3+^) in response to redox changes in the cell [14,18,56].

Electron transfer between Ngb(Fe^2+^) and Cyt *c*(Fe^3+^) was also detected by stopped-flow spectroscopy [57], while recombinant human Ngb was reduced with an excess of sodium dithionite, which most likely led to the disulfide bridge cleavage as well.

Based on the aforementioned data, two hypotheses regarding the interaction between Ngb(Fe^2+^) and Cyt *c*(Fe^3+^) in vivo were formed [30,43,56]. According to one of them [30,43,52], Ngb(Fe^2+^) in the ferrous deoxy form, which normally prevails in cells [15,16,17], reduces Cyt *c*(Fe^3+^) molecules by leaving the mitochondria into the cytoplasm to Cyt *c*(Fe^2+^), thereby preventing apoptosome assembly and apoptosis triggering along the Cyt *c*-dependent pathway (Figure 1, I). It should be noted that such a mechanism takes place under normal cellular conditions, thus preventing the consequences of the “accidental” Cyt *c*(Fe^3+^) release from mitochondria [30], which is a constitutive process associated with transient VDAC pore opening [58,59,60]. Even through “accidental” Cyt *c*(Fe^3+^) release is a small-scale process, Cyt *c*(Fe^3+^) can bind to the inositol-1,4,5-triphosphate receptors on the endoplasmic reticulum, thus causing Ca^2+^ release to the cytoplasm. An increase in the level of cytosolic Ca^2+^ further stimulates Cyt *c*(Fe^3+^) release from the mitochondria, thus establishing an ongoing amplification signal of the initial Cyt *c*(Fe^3+^) release [30,61]. Therefore, the existence of a resetting mechanism for the threshold level of Cyt *c*(Fe^3+^), for instance, via redox reaction with Ngb(Fe^2+^), is highly probable [30]. This hypothesis also explains the predominant localization of Ngb in neurons and retinal cells at high concentrations [2,3]. These highly specialized and metabolically active cells experience frequent high fluxes of cellular Ca^2+^ during their normal physiological functioning. As a result, these cell types might have a tendency to undergo programmed cell death as a result of the “accidental” release of Cyt *c*(Fe^3+^) in the cytoplasm. Hence, higher cellular concentrations of Ngb are required to protect these cells from apoptosis [30]. This hypothesis is further supported by the data on the Ngb localization in close proximity to mitochondria [7]. It is important to note that, in accordance with this mechanism, Ngb can suppress the triggering of the apoptotic process through the “accidental” release of Cyt *c*(Fe^3+^) while still allowing committed programmed cell death to occur under appropriate circumstances [30]. 

According to another hypothesis [56], electron transfer from Ngb(Fe^2+^) to Cyt *c*(Fe^3+^) occurs only in the presence of the disulfide bridge in the Ngb structure, which corresponds to the oxidative stress conditions [14,18,56]. In support of this idea, *NGB* gene expression is upregulated in response to various stress signals [4,5]. Moreover, although Ngb cellular concentration is primarily high only in neurons and retinal cells [2,3], it can increase in other cell types under cellular stress conditions to a level that is sufficient for Ngb-mediated protection against apoptosis [4,5]. This hypothesis contradicts the one described above, but not vice versa. It is quite probable that electron transfer from Ngb(Fe^2+^) to Cyt *c*(Fe^3+^) is possible in both cases: from Ngb(Fe^2+^) with the disulfide bridge and from Ngb(Fe^2+^) with reduced cysteines. In this case, the disulfide bridge may play a crucial role in modulating this electron transfer reaction under the corresponding redox conditions. It should be noted that, under oxidative stress conditions, not only cysteine residues, but also the heme of Ngb can be oxidized, which will lead to the inability of the Ngb to act as electron donor for the Cyt *c*. In addition, a Ngb-reducing system in vivo is still unknown, although several attempts have been made to identify it [43,62,63]. Hence, in the case of Ngb heme oxidation, the interaction with Cyt *c* is either disrupted or follows other mechanisms without electron transfer [54]. 

Thus, the studies on electron transfer between Ngb(Fe^2+^) and Cyt *c*(Fe^3+^) are rather inconsistent and limited. This inconsistency arises from the experimental difficulties, such as the preparation of pure ferrous Ngb without a reducing agent as a consequence of Ngb’s high autoxidation rate (0.23 ± 0.03 min^−1^ [17]). The reducing agent molecules can compete with Ngb(Fe^2+^) for the reduction of Cyt *c*(Fe^3+^) molecules in the solution; thus, the possibility of false positive results originates. Another challenge is the inability to use a classical strategy for the study of the electron transfer between cytochromes and heme globins. This strategy utilizes the difference in CO binding between six-coordinated cytochromes and five-coordinated heme globins, which cannot be used given the six-coordination nature of Ngb [57]. It is evident that further studies of the electron transfer between Ngb(Fe^2+^) and Cyt *c*(Fe^3+^) with alternative approaches are required. 

## 3. Interactions between Neuroglobin and Cytochrome *c* without Electron Transfer

Due to the Ngb(Fe^2+^)-Cyt *c*(Fe^3+^) electron transfer studies being challenging, the interactions between ferric forms of Ngb and Cyt *c* have also been investigated [54,64,65,66]. Various techniques, including isothermal titration calorimetry (ITC), ^1^H NMR, surface plasmon resonance (SPR), and quartz nanopipettes, have been employed to examine the formation of a transient Ngb–Cyt *c* complex. The equilibrium dissociation constant for this complex has been reported to range from 10 to 45 µM [54,64,65,66]. These values are comparable to the known equilibrium dissociation constants for the Cyt *c*-cytochrome–*c* peroxidase redox complex [67,68]. These findings support the formation of a short-lived redox Ngb–Cyt *c* complex [64]. In addition, an increase in the solution ionic strength was found to cause an increase in the equilibrium dissociation constant (by up to 120 µM). This dependence on the solution ionic strength indicates the involvement of electrostatic interactions between proteins in the process of Ngb–Cyt *c* complex formation [64]. Moreover, at a neutral pH, Ngb (pI 4.6) has a negative charge, while Cyt *c* (pI 10.2) has a positive charge, thus further confirming the role of electrostatic interactions in the Ngb–Cyt *c* complex formation [52]. 

It should be noted that the presence of the Ngb disulfide bridge does not affect the stability of the Ngb–Cyt *c* complex, since there are no differences between the equilibrium dissociation constants for Cyt *c* complexes with recombinant human and rat Ngb [66].

Most studies on the interaction between ferric Ngb and Cyt *c* have been focused on the confirmation of the transient Ngb–Cyt *c* complex formation for the electron transfer from Ngb(Fe^2+^) to Cyt *c*(Fe^3+^) rather than the demonstration of a physiologically significant interaction between the ferric forms of these proteins. However, a hypothesis has been put forward that the interaction between Ngb(Fe^3+^) and Cyt *c*(Fe^3+^) may be sufficient to inhibit the initiation of apoptosis through the Cyt *c*-dependent pathway [53] (Figure 1, II). This hypothesis is based on the results obtained by a quantitative model that simulated apoptotic events. It was found that the interaction of Ngb(Fe^3+^) and Cyt *c*(Fe^3+^) could block the intrinsic apoptosis pathway. Nevertheless, the authors emphasized that higher intracellular concentrations of Ngb are required in this case in comparison to the scenario where a redox reaction between Ngb(Fe^2+^) and Cyt *c*(Fe^3+^) occurs. This idea is further supported by the computational data on Cyt *c* Lys72 involvement in the Ngb(Fe^3+^)–Cyt *c*(Fe^3+^) complex formation [54]. Cyt *c* Lys72 is also a key residue involved in the Cyt *c*(Fe^3+^)–Apaf-1 complex formation [46,69]. Therefore, it is suggested that Ngb and Apaf-1 compete for the interaction with Cyt *c*(Fe3+), which may reduce the amount of assembled apoptosomes and interfere with further apoptotic events from triggering.

It should be emphasized that the experimentally determined equilibrium dissociation constant for the Ngb–Cyt *c* complex is five orders of magnitude higher than the equilibrium dissociation constant of the Cyt *c* complex with Apaf-1 (10^−10^ M) [70]. In this regard, if Ngb and Apaf-1 bind competitively to Cyt *c*, the probability of Cyt *c*–Apaf-1 complexes formation is higher. Moreover, in the absence of the electron transfer between Ngb and Cyt *c*, the latter will not lose its proapoptotic properties as a result of this interaction. Therefore, it can be considered unlikely that the interaction between the ferric forms of Ngb and Cyt *c* can effectively prevent apoptosis triggering. However, binding affinities determined using isolated proteins only may not represent the true nature of the interactions in the cellular environment. Several other factors such as binding sites, the quantity of released Cyt *c*, and the cellular concentrations of Ngb and Apaf-1 may play important roles in the protein–protein interactions. Hence, is possible that the interaction between the ferric Ngb and Cyt *c* contributes to Ngb functionality as an anti-apoptotic protein; however, this contribution is likely to be minor. 

Thus, the interaction between Ngb and Cyt *c* appears to be composed of two steps: (1) the convergence of the protein molecules and the formation of the short-lived complex mainly due to electrostatic interactions; (2) electron transfer from Ngb(Fe^2+^) to Cyt *c*(Fe^3+^), followed by the dissociation of the complex. 

## 4. Interactions between Neuroglobin and Cytochrome *c* as a Blockade Mechanism for the Intrinsic Apoptosis Pathway

The interactions between Ngb and Cyt *c* as one of the mechanisms of the Ngb neuroprotective function implies the blockade of apoptosis from triggering through the intrinsic pathway. Hence, the effectiveness of this mechanism can be evaluated by the reduction of programmed cell death events and the decrease in caspase activity. It was demonstrated that the overexpression of the *NGB* gene in the human neuroblastoma cell line SH-SY5Y (with a cellular Ngb concentration of about 5 μM) increased its resistance to apoptosis induced through the inhibition of Bcl-2 proteins in comparison to the wild-type cell line [53]. Of note, the inhibition of Bcl-2 proteins suggests the triggering of apoptosis specifically through the intrinsic pathway. In vitro experiments showed that the addition of Ngb and Cyt *c*(Fe^3+^) to a cell lysate containing major components of the apoptosome (dATP, Apaf-1, and procaspase 9) led to a significant decrease in caspase 9 activity compared to the experiment where only Cyt *c*(Fe^3+^) was added. These findings highlight the neuroprotective function of Ngb in inhibiting the intrinsic pathway of apoptotic cell death [53]. 

Mass spectrometry analysis revealed the interaction between Ngb and Cyt *c* in hypoxic murine neuronal (HN33) cell lysate, while no such interaction was observed in normoxic cell lysate [71].

Despite a large number of experimental and computational studies on the interaction between Ngb and Cyt *c*, the issue of the subcellular localization of this process is not fully understood. Ngb can be localized not only in close proximity to mitochondria, but also within mitochondria [6,7,8]. In addition, upon oxygen–glucose deprivation (OGD), a relative increase in the mitochondrial Ngb concentrations (in comparison with cytosolic concentrations) occurs in primary cultured murine cortical neurons [39] and the SH-SY5Y cells [72]. OGD leads to triggering apoptosis through the Cyt *c*-dependent pathway. Moreover, *NGB* gene overexpression under OGD led to a decrease in the cytosolic Cyt *c* concentrations (compared to the wild-type cell line), which indicates the inhibition of Cyt *c* release from mitochondria [39,72]. On the one hand, these results can be explained by Ngb interaction with VDACs (Figure 1, III) [39]. On the other hand, it is possible that the Ngb–Cyt *c* interaction in the intermembrane space is also causative for decreased Cyt *c* release from mitochondria (Figure 1, I and II). In support of the latter, after the OGD of SH-SY5Y cells, the colocalization of Ngb and Cyt *c* in the mitochondria was demonstrated [72].

The relocation of Ngb to mitochondria under the influence of the sex steroid hormone 17β-estradiol was also reported during H_2_O_2_-induced apoptosis in human neuroblastoma SK-N-BE cells. The association of Ngb and Cyt *c* within the mitochondria and a decrease in cytosolic Cyt *c* levels, along with a decrease in caspase 3 activity, was observed under such conditions [8].

Therefore, the interaction between Ngb and Cyt *c* can occur not only in the cytoplasm in close proximity to mitochondria, but also in the mitochondrial intermembrane space, thereby preventing the release of Cyt *c* from the mitochondria. It is worth noting that this localization of the Ngb–Cyt *c* interaction makes it possible to get away from the competition for binding to Apaf-1 in the cytoplasm.

## 5. Modeling of the Neuroglobin–Cytochrome *c* Complex

Redox protein complexes are characterized by relatively high dissociation constants and, accordingly, exist for a short time. Hence, the isolation and structure determination of such complexes are challenging. Although many intermolecular electron transfer processes have been identified, very few redox complexes have been isolated, nor have their structures been determined. In this regard, computational methods are widely used to identify the putative structures for redox protein complexes.

In particular, molecular docking was employed to model the Ngb–Cyt *c* complex structure, as reported in several publications [54,64,66,73,74]. Additionally, Alphafold2 prediction [75] and molecular dynamics [54,74] were utilized. In some studies, computational methods were combined with experimental techniques such as SPR, ITC, and stopped-flow spectroscopy. Specifically, interactions between Ngb mutant variants or peptides of the Ngb sequence (with and without mutations) and wild-type Cyt *c*, as well as vice versa, were investigated [54,57,75]. This approach enabled the identification of amino acid residues from the protein–protein interaction surface. The results of these studies are presented in Table 1.

Ngb amino acid residues presumably involved in their interaction with Cyt *c* belong to E and F α-helices surrounding the heme group (Figure 2). Most of the Cyt *c* amino acid residues from the putative interaction surface with Ngb belong to either the Cyt *c* universal binding site, which is a cluster of surface Lys residues located around the heme cavity, or the red Ω loop (70–85).

The findings from the binding surfaces modeling and the identification of key amino acid residues on the surfaces of Ngb and Cyt *c* support the notion that electrostatic interactions are crucial for the transient Ngb–Cyt *c* complex formation, as reported previously [64]. Particularly, the negatively charged Ngb amino acid residues Glu60, Asp63, Asp73, and Glu87, as well as the positively charged Cyt *c* amino acid residues Lys25, Lys27, and Lys72, are involved in the interaction [54,57,64,66,73,75]. Notably, Cyt *c* Lys72 is the key residue involved in the formation of the Cyt *c*(Fe^3+^) complex with Apaf-1, with the peripheral residue Lys25 also contributing to the complex formation [46,69]. Thus, the involvement of these Cyt *c* amino acid residues in the formation of the transient Ngb–Cyt *c* complex may suggest the competitive inhibition of Cyt *c*(Fe^3+^)–Apaf-1 complex formation by Ngb. 

Furthermore, hydrophobic interactions, involving the Ile81 of Cyt *c* [75], and hydrogen bonds, such as Thr77(Ngb)–Lys7(Cyt *c*) [54] and Val99(Ngb)–Gln16(Cyt *c*) [74], are also likely to contribute to the Ngb–Cyt *c* complex stabilization.

Thus, the putative Ngb–Cyt *c* interaction surface has been determined by a combination of computational and empirical methods. However, further studies are required to clarify the actual Ngb–Cyt *c* interaction surface, which include such studies as those on the interactions between Ngb and Cyt *c* using mutant protein variants. To date, only a few studies have investigated the contribution of individual Ngb residues to the formation of the Ngb/Cyt *c* complex using mutant Ngb variants [57] or peptides of the Ngb amino acid sequence (with and without mutations) [54]. Additionally, the involvement of only one residue (Ile81) of Cyt *c* in the transient Ngb–Cyt *c* complex formation has been investigated [75]. The determination of the Ngb–Cyt *c* interaction surface, as well as identifying the key residues involved in this interaction, are necessary for understanding the mechanisms underlying the neuroprotective function of Ngb.

## 6. Conclusions

This review presents a comprehensive summary of the available data on the molecular interactions between Ngb and Cyt *c*, which are two heme proteins. The existing body of evidence strongly supports the in vivo interaction between Ngb and Cyt *c*, which represents at least one mechanism underlying the neuroprotective function of Ngb. However, further investigations are necessary to elucidate the structure of the Ngb–Cyt *c* redox complex, as well as to clarify the importance of the electron transfer reaction for effective neuroprotection. Additionally, the role of the Ngb disulfide bridge in modulating its interaction with Cyt *c*, which involves the subcellular localization of this interaction, as well as the contribution of this mechanism to the neuroprotective function of Ngb in comparison to other known mechanisms, require further exploration. Comprehension of the mechanisms underlying Ngb-mediated neuroprotection is crucial for advancing toward potential practical applications, including the rational design of Ngb-based drugs aimed at the therapy of various diseases associated with the apoptotic death of neuronal cells.

## Figures and Tables

**Figure 1 biomolecules-13-01233-f001:**
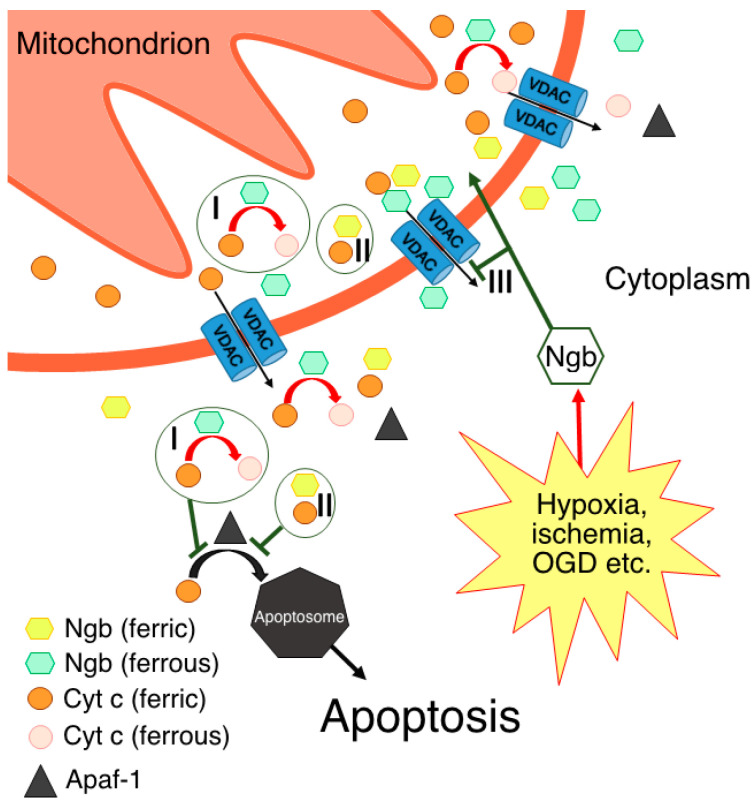
Possible molecular mechanisms of Ngb-mediated neuroprotection through direct or indirect interactions with Cyt *c*: I—electron transfer reaction, thus resulting in Cyt *c* being unable to form apoptosome via its interaction with Apaf-1; II—formation of complex between ferric forms of Ngb and Cyt *c*; III—prevention of Cyt *c* release in the cytoplasm via Ngb interaction with VDACs.

**Figure 2 biomolecules-13-01233-f002:**
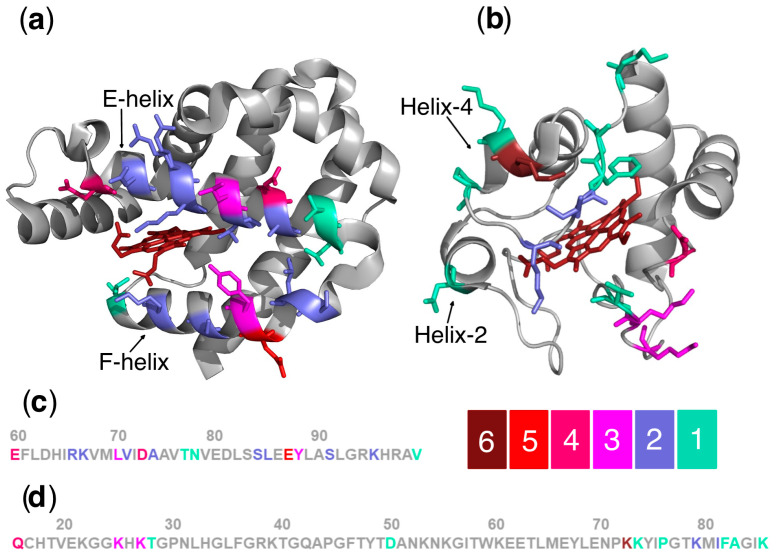
Putative interaction surface of Ngb (**a**) (ID PDB: 4MPM [11]) and Cyt *c* (**b**) (ID PDB: 1HRC [76]). The amino acid residues of Ngb and Cyt *c* are shown coded in colors according to the scale of the image. It expresses the number of publications where these amino acid residues were indicated to be involved in the Ngb–Cyt *c* interaction. Parts of the protein sequence of Ngb (**c**) and Cyt *c* (**d**) with these amino acids are also shown. The image was generated with PyMOL program.

**Table 1 biomolecules-13-01233-t001:** Amino acid residues involved in the formation of the Ngb–Cyt *c* complex. Several residues in one protein that establish interfacial contact with another protein are separated by a comma on the same rows. The residues identified in three or more studies are in bold.

Ngb Residues	Cyt *c* Residues	Methods	Ref.
**E60**	**K72**	Molecular docking (BiGGER)	[64]
**E87**	**K25**		
R66	D50		
**E60**	**K72**	Molecular docking (ZDOCK, PatchDock, GRAMM-X, PDBePISA)	[73]
**E87**	**K25**		
**E87**	**K27**		
S84	P76	Molecular docking (ZDOCK)	[66]
**D73**	K73		
heme	**K72**		
R66	**Q16**		
heme	K79		
V99	**Q16**	Molecular docking (ZDOCK), molecular dynamics (GROMACS)	[74]
**E87**, **Y88**, S91	**Q16**	Molecular docking (ZDOCK), molecular dynamics (NAMD)	[54]
**E87**	**K27**		
S84	T28		
**L70**, **D73**, T77	**K72**		
T77, N78	K79		
K67, **L70**, V71, A74, L85, **Y88**, heme	I81		
**L70**	F82		
K67, **L70**	A83		
L85, **Y88**	heme		
**D73**	**K72**	Molecular docking (ZDOCK), molecular dynamics (NAMD), SPR (Cyt *c* and peptides of the Ngb sequence (with and without mutations)	
**E87**	**K27**		
T77	**K72**		
**D73**, D63, **E60**, **E87**		Stopped-flow spectroscopy (Cyt *c* and Ngb mutants)	[57]
**L70**, **Y88**, K67, K95		Molecular docking (ZDOCK) based on stopped-flow spectroscopy data	
	**K25**, **K27**, **K72**		
**D73**	**K72**	Alphafold2 prediction	[75]
D63	K86		
S91	**Q16**		
K95	**Q16**		
**L70**, V71, A74, L85, **Y88**	I81	Alphafold2 prediction, ITC (Ngb and mutant Cyt *c*)	

## Data Availability

This review systematized and analyzed publicly available and previously published research data.

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
