# Peer review of "Molecular Interactions between Neuroglobin and Cytochrome *c:* Possible Mechanisms of Antiapoptotic Defense in Neuronal Cells"

_biomolecules, 2023, doi:10.3390/biom13081233_

Round 1
Reviewer 1 Report
Please check the english grammar. For example line 13- neuronal survivor (survival)
line 24-neuroglobin is a six coordinate....
line 47- Similar to hemoglobin and myoglobin (,) Ngb reversibly
Please arrange the numeration of the chapters 4 through 6 (pages 5-9).
Check the abreviation of neuroglobin. Sometimes is spelled as NGB , the NGB or Ngb.
Is Figure 2 (interaction surface of NgB), an original figure for this review?
Please check the english grammar through the text.
Author Response
We thank the Reviewer for the careful reading of the manuscript and valuable comments. With Reviewer’s permission, we give answers to comments step by step.
- Please check the english grammar. For example line 13- neuronal survivor (survival)
line 24-neuroglobin is a six coordinate....
line 47- Similar to hemoglobin and myoglobin (,) Ngb reversibly
Answer. English grammar was checked and corrected through the text.
- Please arrange the numeration of the chapters 4 through 6 (pages 5-9).
Answer. Done.
- Check the abbreviation of neuroglobin. Sometimes is spelled as NGB , the NGB or Ngb.
Answer. Checked and corrected. Ngb is used when we write about the protein, NGB is used when we write about the gene. We added the word "gene" after NGB to avoid confusion.
- Is Figure 2 (interaction surface of NgB), an original figure for this review?
Answer. Yes, Figures 1 and 2 are original figures that have not been published before.
Reviewer 2 Report
This review is a helpful, detailed overview of the role of the neuroglobin interactions with cytochrome c and its regulation of apoptosis triggered by cytochrome c release into the cytosol.
Table 1 is an useful overview of the predicted interaction surface between the two proteins. The one topic that was not addressed, is the impact of any of the known post-translational modification of cytochrome c, which is phosphorylated, and or acetylated, at multiple residues in different tissues, including brain cells.
The paper is clear and understandable. However, I suggest the authors consider the following alternative phrases (by line number).
16. with and without > with or without
17. is aimed to provide > provides
29. while > although
33. displays > has a
39. of > of the
41. bridge > bridge,
42. strengthen > strengthened
42. in > in a
44. a necessary and [delete]
46. without > without a
50. survivor > survival
85. of > of the
90. both under factors and > under
223. implies > requires
223. through the
225. and the decrease in caspase activity
226. NGB > NGB gene
228. to > to the
259. acids > acid
192. is obtained by the PyMOL program. > was generated with PyMOL.
Author Response
This review is a helpful, detailed overview of the role of the neuroglobin interactions with cytochrome c and its regulation of apoptosis triggered by cytochrome c release into the cytosol.
Table 1 is an useful overview of the predicted interaction surface between the two proteins.
We thank the Reviewer for the careful reading of the manuscript and valuable comments. With Reviewer’s permission, we give answers to comments step by step.
- The one topic that was not addressed, is the impact of any of the known post-translational modification of cytochrome c, which is phosphorylated, and or acetylated, at multiple residues in different tissues, including brain cells.
Answer. This topic indeed was not addressed since in reviewed literature this topic was not brought out. However, we suppose that post-translational modifications of Cyt c do not affect the Ngb-Cyt c interaction significantly since in different studies different Cyt c were used: recombinant Cyt c (bacterial expression system which means there are no post-translational modifications), Cyt c from horse heart, Cyt c in different cell lines.
- The paper is clear and understandable. However, I suggest the authors consider the following alternative phrases (by line number).
- with and without > with or without
- is aimed to provide > provides
- while > although
- displays > has a
- of > of the
- bridge > bridge,
- strengthen > strengthened
- in > in a
- a necessary and [delete]
- without > without a
- survivor > survival
- of > of the
- both under factors and > under
- implies > requires
- through the
- and the decrease in caspase activity
- NGB > NGB gene
- to > to the
- acids > acid
- is obtained by the PyMOL program. > was generated with PyMOL.
Answer. Corrected, except line 223 (highlighted with green) because it will change the meaning of the sentence.
Reviewer 3 Report
In this review by Semenova et al., authors have detailed interactions of neuroglobin with its known binding partner cytochrome c. This review has a compilation of different studies so that reader can obtain a good picture of the interactions in this review. I recommend this review for publication in Biomolecules after addressing following comments:
Comments to Authors:
1) In line# 109, the results are mentioned in consistent with previous results but in the line#101, it is mentioned “somewhat controversial”. Authors should clearly mention how the controversy has been established based on a general consensus in prior results and the results in Ref#56. This allows readers to have a clear idea of what authors are trying to convey.
2) In Figure 1, Authors mentioned the electron transfer reactions but there is no formal reaction is texted. I suggest Authors to add this part for broader readership.
3) In line#212 few lines (…it is unlikely that the interaction between the ferric forms of Ngb and Cyt c can effectively prevent the apoptosis triggering…) are not supported with further validations and/or evidences. Therefore, the conclusions based on the binding affinity of isolated proteins only may not represent the true nature of interactions in the cellular environment. Several other factors such as binding sites, quantity of released Cyt c play important role.
4) I advise authors to doublecheck proper abbreviation of journal names before resubmission. For example, the journal name in the Ref#8 should be abbreviated as Cell Death Dis.
5) The Figure captions (Figure 1) do not show any information whether the figures have been adapted from prior publications with proper authorizations to use. I suggest authors to work on this with proper citation, if needed.
Author Response
In this review by Semenova et al., authors have detailed interactions of neuroglobin with its known binding partner cytochrome c. This review has a compilation of different studies so that reader can obtain a good picture of the interactions in this review. I recommend this review for publication in Biomolecules after addressing following comments.
We thank the Reviewer for the careful reading of the manuscript and valuable comments. With Reviewer’s permission, we give answers to comments step by step.
1) In line# 109, the results are mentioned in consistent with previous results but in the line#101, it is mentioned “somewhat controversial”. Authors should clearly mention how the controversy has been established based on a general consensus in prior results and the results in Ref#56. This allows readers to have a clear idea of what authors are trying to convey.
Answer. We have corrected this part to make the contradiction between ref43 and ref56 clearer.
2) In Figure 1, Authors mentioned the electron transfer reactions but there is no formal reaction is texted. I suggest Authors to add this part for broader readership.
Answer. Formal electron transfer reaction is added.
3) In line#212 few lines (…it is unlikely that the interaction between the ferric forms of Ngb and Cyt c can effectively prevent the apoptosis triggering…) are not supported with further validations and/or evidences. Therefore, the conclusions based on the binding affinity of isolated proteins only may not represent the true nature of interactions in the cellular environment. Several other factors such as binding sites, quantity of released Cyt c play important role.
Answer. We have corrected this part and have added suggested limitations for the aforementioned statement.
4) I advise authors to doublecheck proper abbreviation of journal names before resubmission. For example, the journal name in the Ref#8 should be abbreviated as Cell Death Dis.
Answer. Done.
5) The Figure captions (Figure 1) do not show any information whether the figures have been adapted from prior publications with proper authorizations to use. I suggest authors to work on this with proper citation, if needed.
Answer. Figures 1 and 2 are original figures that have not been published before.
Reviewer 4 Report
Authors present a comprehensive summary of the available data on the molecular interactions between neuroglobin (Ngb) and mitochondrial cytochrome c (Cyt c), two heme proteins. Since Ngb has been shown to promote neuronal survivor in conditions such as ischemia, hypoxia, Alzheimer's and Huntington's diseases, oxidative stress, etc., Authors hypothesize that the interaction between Ngb and Cyt c may serve as at least one of the mechanisms of NGB-mediated neuroprotection. The review is well written, well organized and all hypotheses are sustained by up-to-date literature. Some minor description may be taken into consideration from Authors to enlarge the interest of the audience.
Line 130-137. Authors explain the first hypothesis underlying the existence of a resetting mechanism for the threshold level of Cyt c(Fe3+) via redox reaction with Ngb(Fe2+). They stated that this hypothesis could also explain the predominant localization of Ngb in neurons and retinal cells at high concentrations concluding that higher cellular concentrations of Ngb are required to protect these cells from apoptosis. However, Ngb has been described as an inducible compensatory protein which levels could be modulated also by stress. This referee suggests adding few sentence on the modulation of Ngb levels to avoid the idea that the interaction between Ngb and Cyt c could happen just in these cell lines.
Always in the same paragraph, it could be interesting for the reader to have some indication about the time necessary for the diverse interactions between Ngb and Cyt c (i.e., Electron transfer between ferrous neuroglobin and ferric cytochrome c versus Interactions between neuroglobin and cytochrome c without electron transfer)
Finally, in the conclusions few sentences on the impact of the interaction between Ngb and Cyt c as one of the mechanisms of NGB-mediated neuroprotection should be added
Author Response
Authors present a comprehensive summary of the available data on the molecular interactions between neuroglobin (Ngb) and mitochondrial cytochrome c (Cyt c), two heme proteins. Since Ngb has been shown to promote neuronal survivor in conditions such as ischemia, hypoxia, Alzheimer's and Huntington's diseases, oxidative stress, etc., Authors hypothesize that the interaction between Ngb and Cyt c may serve as at least one of the mechanisms of NGB-mediated neuroprotection. The review is well written, well organized and all hypotheses are sustained by up-to-date literature. Some minor description may be taken into consideration from Authors to enlarge the interest of the audience.
We thank the Reviewer for the careful reading of the manuscript and valuable comments. With Reviewer’s permission, we give answers to comments step by step.
- Line 130-137. Authors explain the first hypothesis underlying the existence of a resetting mechanism for the threshold level of Cyt c(Fe3+) via redox reaction with Ngb(Fe2+). They stated that this hypothesis could also explain the predominant localization of Ngb in neurons and retinal cells at high concentrations concluding that higher cellular concentrations of Ngb are required to protect these cells from apoptosis. However, Ngb has been described as an inducible compensatory protein which levels could be modulated also by stress. This referee suggests adding few sentence on the modulation of Ngb levels to avoid the idea that the interaction between Ngb and Cyt c could happen just in these cell lines.
Answer. We have added few sentences about the NGB gene upregulation in response to various stress factors.
- Always in the same paragraph, it could be interesting for the reader to have some indication about the time necessary for the diverse interactions between Ngb and Cyt c (i.e., Electron transfer between ferrous neuroglobin and ferric cytochrome c versus Interactions between neuroglobin and cytochrome c without electron transfer).
Answer. Instead of time we provide the second-order rate constant for the electron transfer reaction (k = 2×107 M-1 s-1) and the equilibrium dissociation constant for the Ngb-Cyt c complex without electron transfer (10-45 µM), which is more accurate than time.
- Finally, in the conclusions few sentences on the impact of the interaction between Ngb and Cyt c as one of the mechanisms of NGB-mediated neuroprotection should be added
Answer. This topic requires further exploration as we have stated in the conclusions “…the contribution of this mechanism to the neuroprotective function of Ngb in comparison to other known mechanisms require further exploration.” Different mechanisms of Ngb-mediated neuroprotection have been studied separately that is why it’s difficult to address the contribution of each.
Round 2
Reviewer 3 Report
I recommend this revised version of the review for publication in Biomolecules after addressing the following minor comment:
Comments to Authors:
1) The sentence in line#225 of the revised version is suggested to revise as below:
……..can effectively prevent the apoptosis triggering. However, binding affinities determined using isolated proteins only may not represent the true nature of interactions in the cellular environment. Several other factors such as binding sites, quantity of released Cyt c and cellular concentrations of Ngb and Apaf-1 may play important role in the the interprotein interactions. Hence,………
Author Response
I recommend this revised version of the review for publication in Biomolecules after addressing the following minor comment:
We thank the Reviewer for the careful reading of the manuscript and valuable comment.
1) The sentence in line#225 of the revised version is suggested to revise as below:
……..can effectively prevent the apoptosis triggering. However, binding affinities determined using isolated proteins only may not represent the true nature of interactions in the cellular environment. Several other factors such as binding sites, quantity of released Cyt c and cellular concentrations of Ngb and Apaf-1 may play important role in the the interprotein interactions. Hence,………
Answer. Done.